# Towards Understanding Excited-State Properties of Organic Molecules Using Time-Resolved Soft X-ray Absorption Spectroscopy

**DOI:** 10.3390/ijms222413463

**Published:** 2021-12-15

**Authors:** Holger Stiel, Julia Braenzel, Adrian Jonas, Richard Gnewkow, Lisa Theresa Glöggler, Denny Sommer, Thomas Krist, Alexei Erko, Johannes Tümmler, Ioanna Mantouvalou

**Affiliations:** 1Berlin Laboratory for Innovative X-ray Technologies (BLiX), D-10623 Berlin, Germany; braenzel@mbi-berlin.de (J.B.); A.Jonas@tu-berlin.de (A.J.); richard.gnewkow@googlemail.com (R.G.); lisa.gloeggler@mailbox.tu-berlin.de (L.T.G.); tuemmler@mbi-berlin.de (J.T.); ioanna.mantouvalou@helmholtz-berlin.de (I.M.); 2Max-Born-Institut für Nichtlineare Optik und Kurzzeitspektroskopie, D-12489 Berlin, Germany; Denny.Sommer@mbi-berlin.de; 3Analytical X-ray Physics, TU Berlin, D-10623 Berlin, Germany; 4Helmholtz Zentrum Berlin, D-12489 Berlin, Germany; 5NOB Nano Optics Berlin GmbH, D-10627 Berlin, Germany; info@nanooptics-berlin.com; 6IAP eV, D-12489 Berlin, Germany; erko@iap-adlershof.de

**Keywords:** NEXAFS, pump-probe, porphyrin, ultrafast X-ray absorption, pseudoisocyanine, TD-DFT

## Abstract

The extension of the pump-probe approach known from UV/VIS spectroscopy to very short wavelengths together with advanced simulation techniques allows a detailed analysis of excited-state dynamics in organic molecules or biomolecular structures on a nanosecond to femtosecond time level. Optical pump soft X-ray probe spectroscopy is a relatively new approach to detect and characterize optically dark states in organic molecules, exciton dynamics or transient ligand-to-metal charge transfer states. In this paper, we describe two experimental setups for transient soft X-ray absorption spectroscopy based on an LPP emitting picosecond and sub-nanosecond soft X-ray pulses in the photon energy range between 50 and 1500 eV. We apply these setups for near-edge X-ray absorption fine structure (NEXAFS) investigations of thin films of a metal-free porphyrin, an aggregate forming carbocyanine and a nickel oxide molecule. NEXAFS investigations have been carried out at the carbon, nitrogen and oxygen K-edge as well as on the Ni L-edge. From time-resolved NEXAFS carbon, K-edge measurements of the metal-free porphyrin first insights into a long-lived trap state are gained. Our findings are discussed and compared with density functional theory calculations.

## 1. Introduction

Sir George Porter stated in his Nobel prize lecture [1] that “… since each molecule has only one ground state, but several excited states, it is clear that this field of investigation is, in principle, a bigger subject than the whole of conventional chemistry …” This statement was based on his work on flash photolysis [2] using flash lamps as well as first available lasers with pulse durations in the nanosecond range. Now, more than 50 years later, femtochemistry [3] is a well-established technology to prepare and control excited-state species on a very fast time scale using ultrashort femtosecond laser pulses. In this regard, pump-probe spectroscopy using a strong optical pump pulse for the preparation of the excited state and a weaker pulse for probing this state is the main experimental approach [4]. This approach has been successfully applied to detect transient states in organic molecules, such as carbocyanines [5,6,7,8], porphyrins [9,10,11,12,13,14] and polyenes [15,16,17,18]. Ultrafast optical transient spectroscopy plays an important role in elucidating charge and energy transfer processes in bacterial [18,19,20,21,22] and plant photosynthesis [23,24,25,26] as well as ligand–metal interactions [10,11] in catalysis. 

The extension of the pump-probe approach to very long or very short wavelengths [27] together with advanced simulation techniques [28,29,30] allow a detailed analysis of excited-state dynamics in organic molecules or biomolecular structures on a ns-to-fs time level. Optical pump soft X-ray probe spectroscopy is a relatively new approach to detect and characterize optically dark states in organic molecules [31,32,33,34] exciton dynamics [35,36] or transient ligand-to-metal charge transfer states in metalloporphyrins [37]. 

Optical pump soft X-ray probe spectroscopy can also contribute to understanding structure–function relationships in natural [38] or artificial “molecular machines” [39]. Time-resolved spectroscopic data taken at optical frequencies are not directly related to large molecular structures at an atomic level, whereas time-resolved soft X-ray absorption spectroscopy (tr-XAS) is capable of probing transient structures on an atomic level [39,40,41]. A detailed knowledge of the molecular structure at an atomic level is indispensable, as was shown, e.g., by X-ray diffraction investigations [24,38] for crystallized parts of the photosynthetic apparatus. 

Recent developments on soft X-ray sources, such as high harmonics generation (HHG) [37,42,43] or laser-produced plasma sources (LPP) [34,44,45], open up new opportunities for studying excited-state dynamics in organic molecules on a laboratory scale. In parallel, the above-mentioned improvements in simulation techniques, together with the tremendous increase of computing power, allow understanding the excited-state behavior even of very complex organic molecules in more detail [28]. 

In this paper, we describe two experimental setups for tr-NEXAFS experiments based on an LPP emitting picosecond and sub-nanosecond soft X-ray pulses in the photon energy range between 50 and 1500 eV. As an example, we apply these setups for NEXAFS investigations of thin films of two organic molecules at the carbon and nitrogen K-edge and compare the results with DFT calculations. In addition, we will show that the method is also capable of elucidating the electronic structure of transition metal compounds playing a role, e.g., in metalloporphyrins.

## 2. Results and Discussion

The calculated HOMO and LUMO iso-surfaces of the wave functions of TAP and PIC have been rendered using the software Avogadro with an isosurface value of 0.01. The result is shown in Figure 1. It can be seen that the wave functions (blue = negative, red = positive) are mainly located at the site of the carbon rings and less at the alkyl substituent, which is common for such organic compounds with conjugated systems.

Ground-state NEXAFS measurements of TAP and PIC samples have been performed at the carbon and nitrogen K-edges using the sub-ns LPP source. The measured spectra can be seen in Figure 2. Additionally, the calculated spectra are depicted in grey. The curves of measured and calculated spectra agree well for all edges. Differences in the relative heights of the spectra are partly due to unknown line widths and to the finite number of calculated transitions. The energetic distances of the transitions agree well, which allows an assignment of isolated transitions to the individual atomic groups via the DFT calculation. From the calculation, the individual contributions of the nitrogen atoms to the C and N K-edge NEXAFS spectrum can be derived. 

### 2.1. TAP

The assignment of the features in the carbon K-edge spectrum of TAP is summarized in Table 1. It follows the explanations given in [34]. According to our TD-DFT calculations, the two features A and C located at 283.9 and 285.4 eV respectively belong to the extended πelectron system of the ring. The first peak is separated by 1.5 eV from the second one, which is very similar to the energetic difference between the SORET- and the Q-band in the UV/Vis spectrum (compare. Figure 11). Features B and E originate from carbon atoms that are only bound to other carbon atoms within the pyrrole ring. Feature D arises solely from the butyl substitution. In the nitrogen K-edge spectrum of TAP, the most prominent feature A′ is the strong peak at 398.3 eV. This peak can be assigned to the 1s -> π* LUMO transition on the porphyrin macrocycle. Feature B′ and D′ at 401.5 eV and 403.4 eV, respectively, are mostly generated by the nitrogen atoms within the pyrrole, which are bound to two carbon atoms and one hydrogen atom. As for the carbon K-edge spectrum, the separation between the peaks A′ and B′ fits very well with the energetic difference between SORET- and Q-band. Feature C′ originates from nitrogen atoms inside of the pyrrole ring that show a double bond with a carbon atom.

For the carbon K-edge of TAP, the transient absorption spectra were measured using the sub-nanosecond tr-NEXAFS setup, as already presented in [34]. The sample was excited at the SORET-band with the third harmonic (343 nm) of the laser with a pulse energy of 1 mJ/cm^2^ and a pulse duration of 0.5 ns. The results are shown in Figure 3. The tr-NEXAFS spectrum was taken at several time delays between 0.2 and 43 ns after the excitation. It has been found that these changes remained constant with time delay. Figure 3 shows in blue the averaged differences of all tr-NEXAFS spectra. In order to validate the results and to obtain a reliable value for the uncertainty of the measured absorption difference ΔA, the difference spectrum with negative time delay (purple curve) is shown. Both curves (blue and purple) have been smoothed into 1 eV-bins using a 15-pixel box filter. The estimated uncertainty of the absorption difference is ±5 × 10^−4^. The biggest light-induced change (d2) can be seen between features A and B at feature B (d3) and between feature B and C (d4). While the transient absorption is reduced for the carbon atoms that are bound to nitrogen, the absorption increased at d1, d2 and d4. Because the energy gap between d3 and d1 matches the energy of the LUMO-HOMO transition, it can be assumed that the density of unoccupied states in the d3 region is transferred to d1. The feature d1 could also possibly arise from an X-ray optical double resonance meaning the 1s electron is transferred to a previously occupied state that is partially depleted due to the laser pulse (cp. Figure 8). The features d2 and d4 cannot easily be isolated due to many possible contributions. However, it can be presumed that the butyl groups do not have an effect on the tr-NEXAFS. The observed slow decay channel (lifetime > 43 ns) responsible for feature d3 could be assigned to a long-living trap state. A similar behavior was recently observed by N K-edge spectroscopy in nonaggregated units in a thin Zn-porphyrin film [46]. To understand the nature of this electronic state, which is unobservable in UV/Vis spectroscopy, in more detail, tr-NEXAFS investigations at the nitrogen K-edge of TAP are planned in the next future.

### 2.2. PIC

The carbon K-edge spectrum of PIC differs significantly from that of TAP. With many chemical similar carbon atoms, the contributions of the different transitions from the C K-edge cannot be easily isolated. Still, it can be seen that the lowest energetic transition (feature F) mainly arises from the carbon atoms that are only bonded to other carbon atoms. Feature G originates from the region around the nitrogen atom, while feature H emerges from the two ethyl groups. Whereas the peak at 283.9 eV is missing, the NEXAFS spectra show two peaks at 285.7 eV and 286.7 eV, separated by only about 1 eV. These structures can be assigned to 1s -> π* transitions, whereas both the bandwidth and position of these features are fingerprints of the aggregation state of the dye [36,47]. In our case, the PIC molecules are only weakly coupled, and they are arranged as H-aggregates in the film (cp. Figure 12). To elucidate the influence of the aggregation state on the NEXAFS spectrum of PIC in more detail, alternative thin film preparation techniques are under investigation.

PIC has two nitrogen atoms that have the same chemical environment. Therefore, both nitrogen atoms contribute equally to the NEXAFS. The three features F′, G′ and H′ can be assigned to transitions of the N 1s electron into different LUMO states. As for the TAP molecule, the energetic difference between the F’ and G’ feature is close to the difference of the peak position of the monomeric first (2.3 eV) and second (3.8 eV) excited-state absorption (cp. Figure 12).

### 2.3. NiO

Figure 4a shows the NiO oxygen K-edge NEXAFS spectrum recorded with the sub-ns LPP together with a spectrum obtained at the synchrotron beamline of PTB (BESSY II-HZB). As can be seen from Figure 4a, the NEXAFS spectrum obtained using the LPP agrees with the related synchrotron data very well, demonstrating the potential of our lab-based NEXAFS. The O K-edge NEXAFS reflects the unoccupied orbital states of the Ni cation that hybridize with the oxygen 2p orbital [48]. There are five main features (A–E) in the NEXAFS spectrum that could be assigned to different electronic states of the molecule (cp. Table 2).

The lowest-energy feature A at 531.6 eV can be attributed to a Ni 3d^8^ state while the features B and C at 537.1 eV and 540.0 eV could be assigned to the Ni 4sp state [48]. The features D–F at higher photon energies are mainly due to multiple-scattering effects of the p photoelectron.

L-edge NEXAFS of Ni as a 3d transition metal mainly corresponds to electric dipole transitions from Ni 2p core levels to 3d orbitals. Due to the strong correlation between the 3d electrons, multiplet structures in the NEXAFS spectrum are predicted by theoretical models [49] using the multi-electron approach. In order to compare these predictions with experimental data in more detail, our measured nickel L-edge NEXAFS spectra were evaluated (Figure 4b). Due to the required high spectral resolution, two different spectrometer settings using the planar RZP in the ns-LPP setup were applied. In order to resolve the L3-edge fine structure (features A′ and B′ in Figure 4b), the RZP on the plane substrate was optimally aligned for the 853 eV region according to a procedure described in [50]. The features C′ and D′ (“multiplet splitting”) could be optimally resolved by alignment of the RZP at 870 eV (L2-edge). For comparison, the spectrum recorded with our proto-type RZP (A9, see above) on a bent substrate is also shown. For this measurement, only 100 images each accumulating 2 laser shots of the ps-LPP were processed. The comparable lower statistical amount is close to resolving the line splitting of Ni L2 and feature A′. The assignments of the features following theoretical predictions and experimental data [49,51] are summarized in Table 2. 

## 3. Materials and Methods

### 3.1. Experimental

The two NEXAFS setups are transmission instruments, where polychromatic soft X-rays from a laser-produced plasma source (LPP) are first transmitted through a thin homogeneous sample and successively dispersed with the help of reflection zone plates. The absorption spectra are then collected with a soft X-ray CCD camera as a detector, see, for example, Figure 5. The individual components are described and discussed in the following chapters.

#### 3.1.1. Laser Produced Plasma Sources

In order to generate soft X-ray radiation in the laboratory, we rely on LPP sources. An intense short laser pulse hits a solid target in vacuum and creates a plasma that emits soft X-ray radiation (SXR). In order to meet different requirements concerning achievable time and spectral structure of the SXR radiation, we have developed two LPP sources enabling state-of-the-art NEXAFS spectroscopy in the laboratory. 

The first LPP source we want to present is pumped by a chirped pulse amplification (CPA) thin-disk laser system. The driving laser operates at 1030 nm with a pulse energy of 120 mJ, a 1.2 ps pulse duration and 100 Hz repetition rate. The plasma is created by focusing the laser pulse on a rotating metal cylinder target. The spot size amounts to 17 μm (FWHM) in diameter, delivering an intensity of the target in about 10^16^ W/cm^2^. This high intensity on the metal target creates a plasma that emits an X-ray spectrum ranging from 50 eV to 1500 eV. Depending on the used target material, the spectrum consists either of characteristic line emission (e.g., Cu, Fe, Sn) or, for some target elements (e.g., W, Au) with high density of multiplet emission, a quasi-broadband spectrum. The LPP source delivers incoherent soft X-ray (SXR) emission with high photon numbers: >10^12^ photons/s*sr @ 0.1% bandwidth in the range 50–500 eV and 10^11^photons/s*sr @ 0.1% bandwidth in the range 500–1500 eV [52]. The pulse duration of the source depends on the pump laser pulse duration as well as on the target material. For a tungsten target and a 1.2 ps pump laser duration, we have estimated an SXR pulse duration at 700 eV of about 10 ps [53]. 

Both the infrared and the SXR optical path is debris-screened against the ablation of the target material. A 100 μm glass plate is used for screening the vacuum window in the pump laser path, and a 900 nm mylar or parylene foil shields the SXR beamline. Having passed the debris foil, the SXR pulse transmits the sample and is collected by a bent reflection zone plate (RZP A9, see below). The whole setup, including the spectrometer, is depicted in Figure 5.

The second LPP setup with a sub-ns-pulse duration has already been described in detail in previous work [34,45]. In brief, this setup uses an Yb:YAG thin-disk laser system with 200 mJ maximum single-pulse energy, 100 Hz repetition rate, 500 ps pulse duration and a solid target for the plasma formation [54]. For the Cu XX line at 1.1594 nm, a brilliance of >10^10^ ph/(mm^2^ mrad^2^ s line) is reached. Using the other metal targets values of >3 × 10^11^ ph/(mm^2^ mrad^2^ s), a line for the water window range, as well as for the EUV, can be achieved [55].

This source utilizes stabilization mechanisms in order to compensate small rotational and translational movements and local variations in the diameter of the target material. The stabilization results in a reduction in source movement from 60 µm to 13 µm standard deviation with a photon-energy-dependent source diameter between 40 µm and 70 µm.

Both sources cover the whole photon energy region relevant for NEXAFS investigation on organo-metallic compounds, starting from the carbon K-edge up to Mg and Al K-edges, the L-edges of transition metals, as well as M-edges of lanthanides.

For transient experiments, small portions of the laser beam can be used as pump, which are perfectly synchronized to the probe pulse. Both setups offer optical delay lines with variable pump-probe separation (ps setup: 1 ps—1.5 ns, sub-ns setup: 0.5–43 ns) and wavelength using non-linear crystals and dye lasers. Due to the complementary temporal pulse structures of the LPPs, transient NEXAFS spectra covering the temporal range for the pump-probe delay from few picoseconds to tens of ns can be recorded.

#### 3.1.2. Reflection Zone Plate Optics

The dispersive element in tr-NEXAFS using LPP sources is a critical component. Due to the isotropic nature of the emitted radiation, it should collect the largest emission angle possible. In order to resolve very small features in the tr-NEXAFS spectrum, a resolving power E/ΔE up to 1000 is required. Finally, a high efficiency (throughput) of the optics is crucial for reasonable data acquisition times. Off-axis reflection zone plates (RZPs) are two-dimensional laminar grating structures, where the grating lines follow an elliptical shape. They focus and disperse broadband radiation from a point source onto the image plane with high efficiency [56,57]. The design wavelength is concentrated in the focus as the image of the source, while energies smaller and larger than the focus energy are displayed as slightly curved lines in an X-shaped pattern on the detector. The sub-ns NEXAFS setup is equipped with two sets of planar off-axis reflection zone plates (NOB, Nano Optics Berlin GmbH) [45]. 

RZPs on planar substrates suffer from a relatively narrow energy range with high spectral resolution around the design wavelength. To overcome this limitation, a special “misalignment” technique has been proposed tuning the focused wavelength of a planar RZP over a broader spectral range [50,57]. Another option to obtain a high spectral resolution over a wide photon energy range is to fabricate RZP structures on a spherical substrate [58]. In contrast to an RZP on a planar substrate, the new design allows a high spectral resolution of up to ΔE/E = 1000 on the detector for a wide spectral range (design-energy ± 50%) without an energy-dependent spatial and spectral limited focusing. This enables recording a spectrum with retaining one dimension for spatial imaging, similar to commonly used Varied Line-Spaced (VLS) grating. In contrast to a VLS, an RZP on a curved substrate offers a high efficiency up to 25% in a large spectral range covering photon energies of up to 1300 eV [58]. 

We tested a prototype (called RZP A9) of this new generation of RZPs that is written on a curved substrate using electron beam lithography. The RZP (NOB Nano Optics Berlin GmbH) consists of three different structures, each 10 mm in height designed for energy of 250 eV, 500 eV, and 780 eV, respectively. The radius of curvature of the RZP substrate amounts to 54.879 m, and the angle of incidence is 2.13° with a reflection angle of 3.52°. The distances between source and RZP and detector and RZP accounted for 1500 mm and 2500 mm, respectively. On the top and bottom of each structure, the RZP has an additional alignment structure for the design-energy of 966.485 eV. The alignment structure represents a miniaturized planar RZP structure. It serves the general alignment of the device as well as for referencing the design energy of the main structure. The RZP A9 was designed to illuminate about 80% of the area of a 1” CCD chip. As an example, Figure 6 shows two detector images, as seen on the CCD, if one RZP structure has been illuminated by soft X-rays from our ps-LPP source. 

#### 3.1.3. General Considerations for Optical Pump X-ray Probe Experiments on Organic Molecules

In optical pump X-ray probe experiments, molecules are excited with a light pulse. Afterwards, the X-ray spectrum of the excited molecule is detected. By varying the time delay between the two short pulses, the temporal evolution of the system can be investigated. The achievable time resolution is given by the longer pulse duration, which is, in most cases, the X-ray pulse durations. Figure 7 illustrates the pump-probe scheme for a typical organic molecule. The optical pump pulse tuned to a transition between the ground singlet state S_0_ and an excited state of the molecule (S_1_, S_2_, …) creates a vacancy in the highest occupied molecular orbital (HOMO), which is detected by an X-ray pulse tuned to the K- or L- absorption edge of the atom of interest.

For successful tr-NEXAFS, it is necessary to excite a sufficient number of molecules depending on the achievable signal-to-noise ratio (SNR) of the setup and the strength of the transient signal. Compared to measurements in liquids, where rapid sample replenishment is possible, the excitation percentage of molecular thin films is often limited to single digits due to radiation-induced damage and sample evaporation. Typical damage thresholds for thin films of organic molecules on Si_3_N_4_ or SiC membranes are in the range of 1–5 mJ/cm^2^. If evaporation due to sample heating is the dominating factor, the maximum power density is almost independent of the pulse length.

Since the absorption coefficient of the optical pump is usually much higher compared to the X-ray probe pulse the maximum sample thickness is determined by the optical pulse, which leads to optically thin samples for the X-ray pulse. The consequence of the low excitation ratio and optically thin samples for X-rays is a small absorption difference in the range of 10^−2^ to 10^−4^. The SNR needed to detect this transient change can only be achieved by taking multiple measurements. This requires a stable setup and will be discussed in the detectors and data acquisition section.

To determine the damage threshold and excitation ratio before the tr-XAS experiment, non-linear absorption (NLA) measurements can be conducted. In an NLA setup, the absorption of a laser through the sample is detected for increasing laser intensities. When part of the sample is excited during the laser pulse, the transmission becomes non-linear due to ground state bleaching. This allows the estimation of the excited-state fraction and modeling of the energy level scheme using a rate equation system for the population densities and a photon transport equation for the radiation transport through the sample [31].

### 3.2. Materials

In order to evaluate the potential of our lab-based tr-NEXAFS setups, we choose three sample systems, see Figure 8: (i) a metal-free porphyrin, (ii) an aggregate forming carbocyanine and (iii) a nickel oxide sample. The metal-free tetra(tert-butyl)-porphyrazine (TAP) is a typical molecule belonging to the class of tetrapyrroles, which exhibits great application potential in optoelectronics and photovoltaics, as well as pigments in natural or artificial photosynthesis. Pseudoisocyanine (1,1′-Diethyl-2,2′-cyanine iodide, PIC) is a J-aggregate forming carbocyanine. J-aggregates of PIC could be regarded as supramolecular polymers that show exceptional photophysical properties, such as giant dipole transition moments and strong exciton-exciton annihilation. NiO as a typical large bandgap semiconductor is applied in photovoltaic solar cells and as an anode material in lithium battery technology. Ni-porphyrin molecules are promising candidates for organic solar cells. The knowledge of its electronic properties is crucial for optimizing the efficiency of these devices. 

Both tr-NEXAFS setups rely on the detection of transmitted radiation through a sample, where the thickness of the sample determines the contrast at the absorption edge and the number of detected photons. Therefore, large, homogeneous thin samples are required. Typically, organic molecule samples are deposited on 150 nm thin Si_3_N_4_ membranes with a window size of 1 mm × 2 mm or 2 mm × 2 mm via spin coating or evaporation using an effusion cell. The choice of preparation is dependent on the chemical properties of the molecule, such as solvability or evaporation temperature (see Appendix A for details). 

After preparation, the organic samples are also pre-characterized to NLA measurements using UV/Vis spectroscopy, an EUV spectrometer for thickness determination and an atomic force microscope (AFM). 

Details concerning instrumentation and measurement parameters can be found in the Appendix A.

### 3.3. NiO

The NiO samples were prepared on 200 nm thick Si_3_N_4_ windows (3 × 3 mm) by depositing Ni using a reactive electron beam evaporation in an oxygen environment at room temperature. The thickness we monitored with a quartz crystal and the stoichiometry with EDX measurements delivered an O:Ni ratio of about 51:48 (see Appendix A). Figure 9 shows the UV/Vis spectrum of a 300 nm thick NiO sample with the bandgap energy of NiO at 3.6 eV [59]. This value is very close to the photon energy of the third harmonic of the pump laser. The optical density (OD = −log(T)) at 3.6 eV accounts for 0.54. In comparison, the related value in the Ni L-edge region (850–870 eV) amounts to 1.1.

### 3.4. TAP

TAP thin films were prepared by evaporation on 1 × 1 mm^2^ Si_3_N_4_ membranes. The UV/Vis spectrum of such a 120 nm thick film in comparison to TAP solved in ethanol (c = 3.6 × 10^−4^ mol/L, 1 mm quartz cuvette) is shown in Figure 11. In porphyrins and its derivatives, the S_1_ state splits into two states due to molecular symmetry, which are named Q_x_ and Q_y_. Transitions to the S_2_ state led to the formation of the Soret band or B band [60].

Figure 10 (inset) shows the thickness distribution using the EUV spectrometer of a 2 mm × 2 mm TAP sample (thickness: 250 nm) after a pump-probe NEXAFS measurement. The pixel values of the CCD images were converted into TAP film thickness using the formalisms explained in [61]. Inhomogeneities in the order of 20 nm to 30 nm can be seen with an average thickness of 250 nm. The relative error of the EUV transmission measurement is estimated to be 20%, and the lateral resolution was 25 μm. The dark dots are small debris particles from the LPP as the sample was not protected by a debris foil in this measurement.

### 3.5. PIC

PIC thin films were also prepared by evaporation. The UV/Vis spectrum of a 250 nm thick PIC sample is depicted in Figure 11. The solution at concentrations below 10^−4^ mol/L in a 1 mm thick quartz cuvette shows absorption bands located at 527 nm, whereas an increase in the concentration leads to the appearance of the characteristic J-band at 573 nm. Our preparation technique results in the formation of a mixture of monomers and H-aggregates in the film with a hypsochromic shift of the H-aggregate absorption band in comparison with the monomer. This behavior is in contrast to other works using a spin coating preparation technique, which results in the formation of J-aggregates [62]. Because in our case the preparation starts with a dye powder rather than an aqueous solution as in [62], the thin film contains mainly H-aggregates and monomers. Simulations of the aggregation process during thin film formation suggest a strong dependence of PIC aggregation on the initial conditions of the preparation process, indicating that, in most cases, a mixture of different aggregates and monomers exits [63].

For the 270 nm thick PIC sample, an AFM image of a 5 µm × 5 µm and a line scan on the edge were performed (cp. Inset in Figure 11). The deviation from the z = 0 position of the AFM tip is shown in the color scale. On the nanometer scale, the film thickness only varies about 1 nm to 2 nm. The line scan at the edge of the sample shows a film thickness of 300 nm ± 10 nm, which agrees with EUV transmission measurements.

### 3.6. DFT Simulations

Calculations based on time-dependent density functional theory (TD-DFT) with the aid of the freeware ORCA [64] were performed to better understand the carbon and nitrogen K-edge absorption spectra. Information from these calculations includes the shape of the HOMO and LUMO, the shape of the NEXAFS spectrum and the assignment of the NEXAFS structures to the involved transitions of the corresponding atoms. For the TD-DFT calculation, the B3LYP functional, together with the def2-TZVP, basis set was used. The RIJCOSX approximation was employed using the auxiliary basis set def2/J. This level of theory was also used to optimize the geometry of the PIC molecule. The geometry of the TAP molecule was optimized by using the universal force field (UFF) method.

TD-DFT calculations deliver oscillator strengths of the first 100 electronic transitions at the respective energy as a delta function for each atom. The delta functions were subsequently convolved with Gaussian functions with a full width at half maximum of 0.5 eV to better reflect the actual shape of the experimental NEXAFS spectrum. The whole spectrum is then uniformly shifted by several eV.

### 3.7. Detectors and Data Acquisition

The standard detectors used are soft X-ray CCD cameras (Greateyes GE 2048 BI) with read-out times of a few hundreds of ms–seconds depending on the number of pixels binned. Alternatively, CMOS detectors offer the possibility for a faster acquisition [44]. 

For the sub-ns setup, the reference and sample spectra are focused with pairs of RZP optics onto the large (27.6 mm × 27.6 mm) sensor area, rendering single-shot NEXAFS spectroscopy feasible. Additionally, a pinhole positioned in the direct path between the source and the detector enables the imaging of the source positions and intensity. NEXAFS measurements using this setup are described in detail in [45]. For the measurements at the carbon K-edge and N K-edge of PIC and TAP samples between 200–20,000 images have been collected. For the NiO O K-edge and NiO Ni L-edge measurements, around 4000 images were processed. For tr-NEXAFS measurements on TAP, the experimental details are summarized in [34].

The ps- setup (cf. Figure 5) with the prototype RZP A9 used full-frame measurements of the 1” CCD detector for either the reference or sample transmitted signal. Hence, the ps-setup with the prototype RZP does not yet allow a coincident reference and measurement signal (cf. Figure 6) nor a faster readout by binning. Concerning the source pointing correction, this setup uses a post-evaluation correction applied on close-to-single laser shot measurement series. Measurements with this setup on the Ni L-edge processed measurement series of about 100 images. 

Further details of the NEXAFS measurements and its evaluation methods are described in [45] and in Appendix A.

## 4. Conclusions and Outlook

We have developed an experimental approach for elucidating the structure of transient electronic states using ultrafast laboratory-based soft X-ray sources. For this purpose, highly brilliant soft X-ray sources were developed, as well as measurement schemes adapted. In order to obtain the required high spectral resolution, advanced X-ray optics with reflection zone plates as dispersive elements were applied. Here, the development of structures on curved substrates enhances the spectral range of high resolving power by a factor of more than 30, rendering efficient analysis of multiple edges feasible. Additionally, EXAFS measurements become easily accessible.

We presented the first measurements with a prototype RZP on a bent substrate in the photon energy range at 850 eV. We could show that this new optical element provides a high spectral resolution for a wide spectral range close to the resolving power of optimized methods for planar RZPs. With a higher number of recorded images, the statistics and by this the resolving power could be improved, favorably with using an sCMOS detector for a significantly shorter read-out time [44]. The current development of this experimental setup focuses on the coincident measurement using two RZPs on curved substrates as on the optimization of alignment routines and post-processing.

With these new laboratory instruments, we pave the way for state-of-the-art tr-NEXAFS spectroscopy independent of large-scale facilities. Thereby, a variety of application fields is opening up, which are by no means restricted to organo-molecular films. The adaption of our approach to other ultrafast lab-based sources, such as harmonics generation [65], is under the way and could extend the time resolution for soft X-ray absorption investigations to the attosecond range [43].

Our TD-DFT calculations carried out for the TAP and PIC films explain the measured NEXAFS spectra well and assign the observed spectral features to the electronic structure of the molecule in its ground state. The simulation of the NEXAFS spectrum of an optically excited state (as for tr-NEXAFS) requires a higher effort (for porphyrin molecules cp. [29]), and it was not in the scope of this publication. However, such simulations are planned for the future. 

Besides the possibility to detect and characterize transient electronic states, our lab-based tr-NEXAFS approach allows the investigation of thermally or non-thermally induced phase transitions as well as relatively slow in situ reactions or slow dynamic processes in catalysis [66]. The application of our setup for quick X-ray absorption fine structure measurements on Ni/TiO_2_ nanostructures is described in [44].

For successful tr-NEXAFS experiments, pre-characterization of samples is a pre-requisite. Through the careful matching of optical pump pulse parameters using UV/Vis spectroscopy and NLA measurements, radiation damage of the sample can be minimized or prevented and transient contrasts enhanced. The design of the spectrometers additionally minimizes radiation damage, as, on the one hand, both the pump and the probe pulse have large footprints on the sample. On the other hand, the systems operate completely background-free, i.e., no residual (probe) light reaches the samples in between pulses. Since the complete spectrum is measured for each laser shot, energy scanning is not necessary, which reduces the radiation dose significantly. Nevertheless, the optical pumping does lead to heating of the samples, thereby possibly introducing temperature effects that complicate the analysis of the transient signals. For this purpose, a sample holder for static temperature measurements is designed for our setups. With this holder, first UV/Vis experiments were conducted, which allow choosing an optimal wavelength for pumping by selecting the wavelength with minimal change in the UV/Vis spectrum. In the future, for each new sample system, a pre-characterization concerning temperature effects will be implemented in the experimental methodology. Since heat loss by thermal radiation is high compared to heat transfer by conduction in nanometer-thin films, the sample holder will use hot gas to heat the sample evenly in the future. A very promising approach to prevent sample damage due to heating effects is the usage of a liquid flat jet carrying the molecule of interest in the solution [67]. However, this approach requires a relatively high volume of sample molecules, not applicable, i.e., for specially prepared parts of photosynthetic apparatus. To overcome this limitation, a liquid cell for transmission measurements on molecules in a solution based on an existing cell for fluorescence experiments [68] is under development.

## Figures and Tables

**Figure 1 ijms-22-13463-f001:**
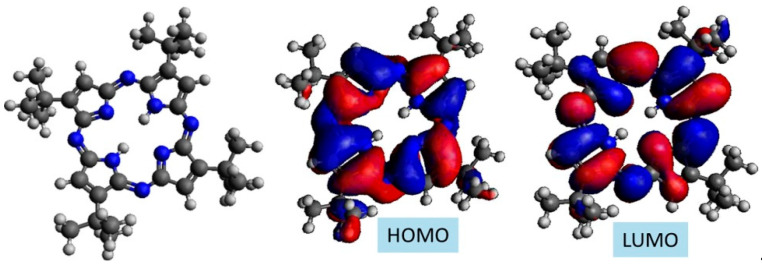
Electronic structure of the HOMO und LUMO of TAP (**top**) and PIC (**bottom**). Carbon atoms are depicted in black nitrogen in blue and hydrogen in white. For both molecules, the HOMO and LUMO extends over the whole conjugated system.

**Figure 2 ijms-22-13463-f002:**
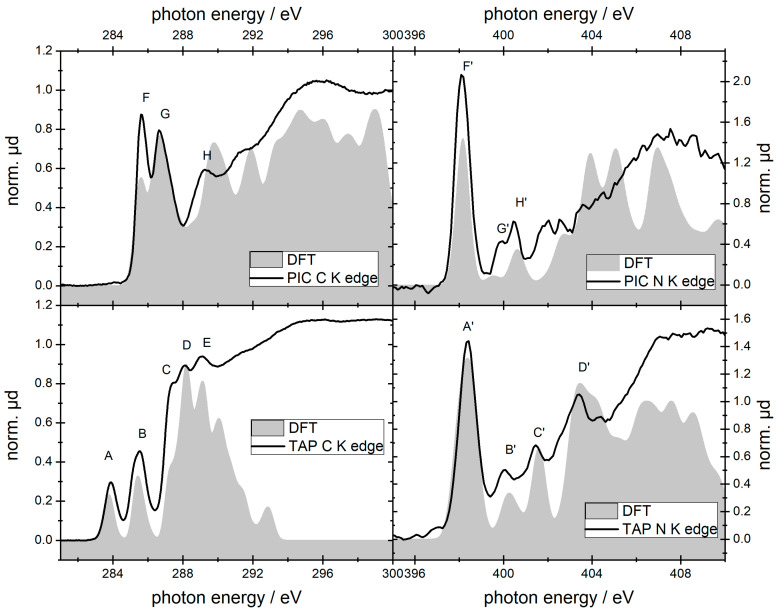
Carbon and nitrogen K-edge NEXAFS spectra of thin films of PIC and TAP in comparison with DFT calculations.

**Figure 3 ijms-22-13463-f003:**
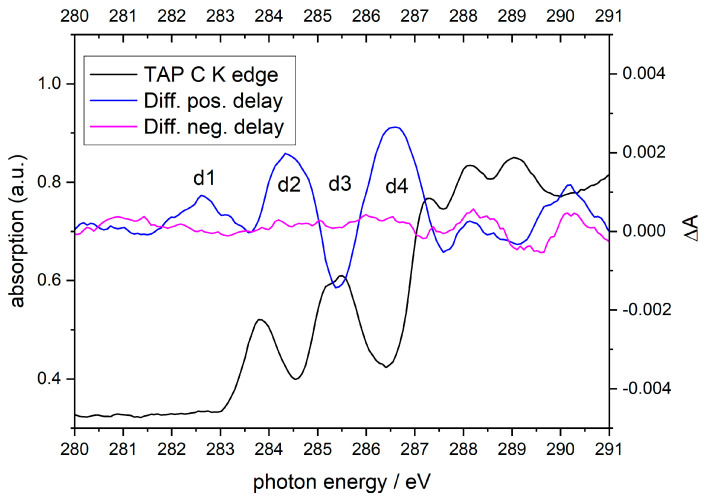
tr-NEXAFS at the carbon K-edge of TAP. The ground state spectrum cf. Figure 2 is shown in black. Die difference spectra with negative and positive time delays between excitation and probing are shown in purple and blue.

**Figure 4 ijms-22-13463-f004:**
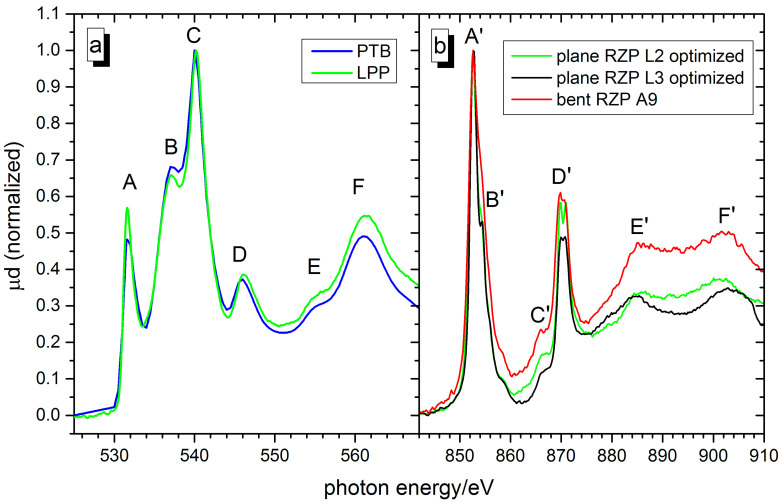
(**a**) Oxygen K-edge NEXAFS spectra of a NiO thin film measured with the sub-ns laser plasma source (green) and at the PTB/BESSY II synchrotron beamline (blue). (**b**) Nickel L-edge NEXAFS spectra of a NiO thin film measured with the sub-ns laser plasma source and an RZP on plane substrate optimally aligned for the L2-edge (black) and L3-edge (red), respectively. For comparison, the spectrum (green) recorded with the ps-LPP and the RZP A9 on a bent substrate is also shown.

**Figure 5 ijms-22-13463-f005:**
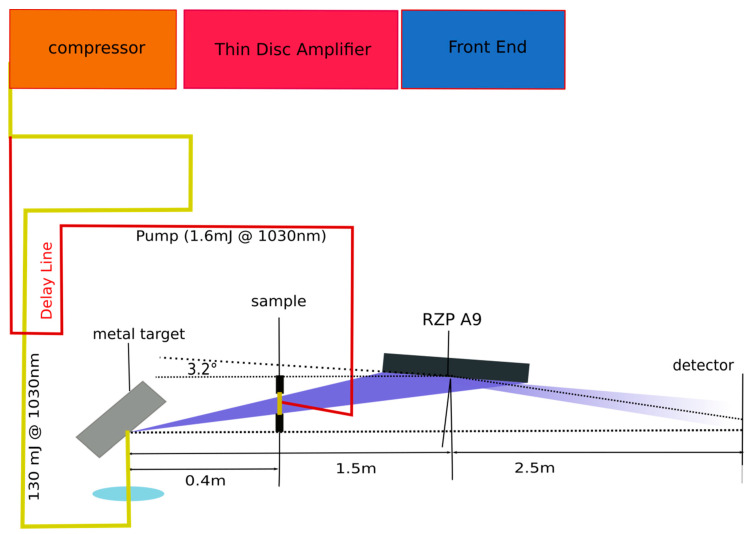
Scheme of the tr-NEXAFS setup using a picosecond driver laser and a reflection zone plate on a bent substrate.

**Figure 6 ijms-22-13463-f006:**
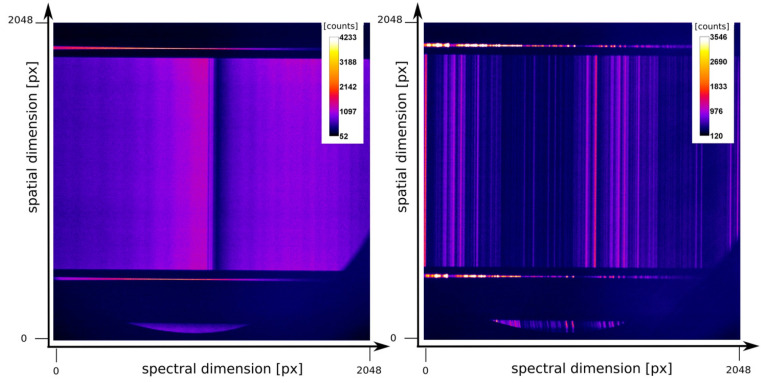
Detector image of a tungsten spectrum (**Left**) taken with the so-called S2 structure (design energy 450 eV) of the RZP A9. In the reference structure on the top and bottom, the oxygen K-edge absorption from the mylar foil is visible. (**Right**): Detector image of an iron spectrum taken with the so-called S3 structure (design energy 780 eV). Design energy of the reference structure: 966.45 eV. The integration time was 945 ms at a laser repetition rate of 100 Hz. Detector: Back-illuminated soft X-ray CCD camera (Greateyes GmbH).

**Figure 7 ijms-22-13463-f007:**
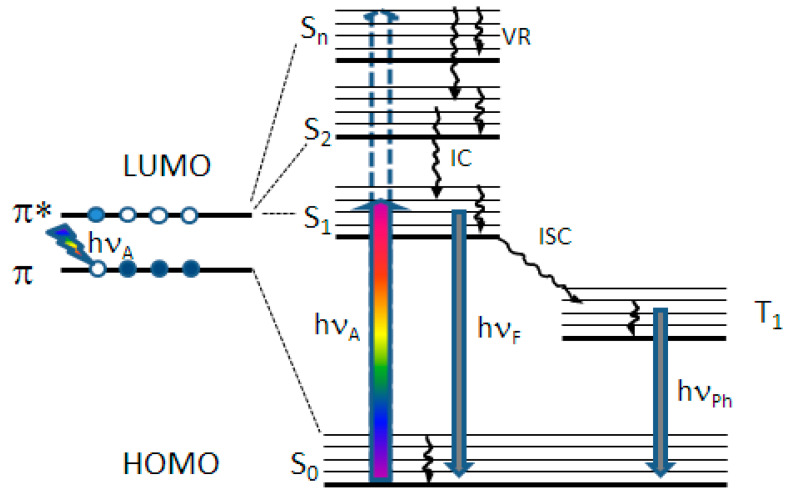
Term scheme illustrating the principle of an optical pump (multi-color arrow, hν_A_) and X-ray probe experiment for a typical organic molecule. The optical pump pulse tuned to a transition between S_0_ and an excited state of the molecule (S_1_, S_2_, …) creates a vacancy in the highest occupied molecular orbital (HOMO), which is detected by an X-ray pulse tuned to the K- or L-absorption edge of the atom of interest. LUMO denotes the lowest unoccupied molecular orbital related to an excited singlet state of the molecule. hν_F_: fluorescence, hν_Ph_: phosphorescence, IC: internal conversion, VR: vibrational relaxation. ISC: intersystem crossing.

**Figure 8 ijms-22-13463-f008:**
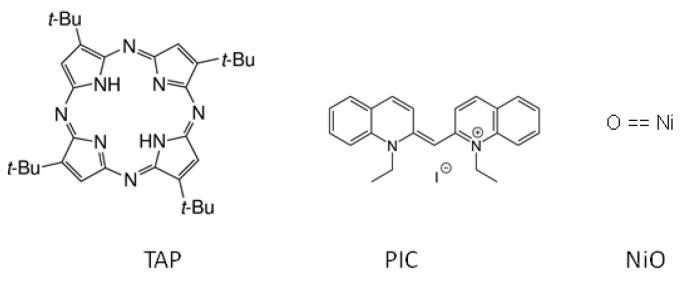
Structure of the three samples under investigation: TAP = tetra(tert-butyl)-porphyrazine), PIC = 1,1′-diethyl-2,2′-cyanine iodide and NiO = nickel oxide.

**Figure 9 ijms-22-13463-f009:**
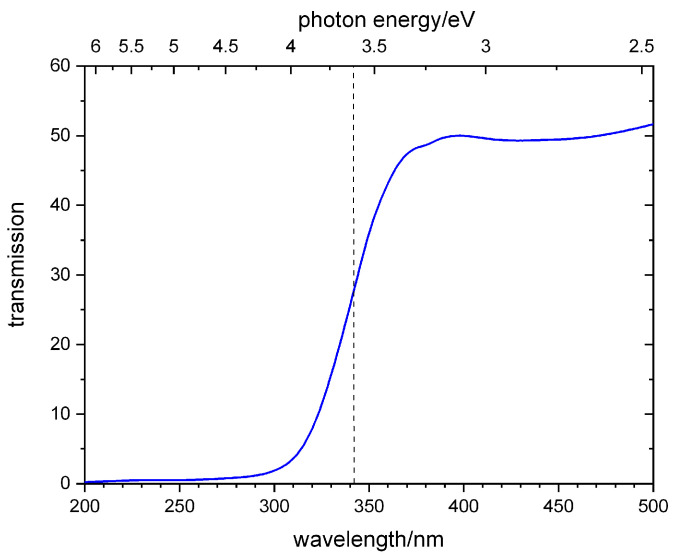
UV/Vis spectrum of the NiO sample (thickness 300 nm) on Si_3_N_4_ corrected for substrate transmission.

**Figure 10 ijms-22-13463-f010:**
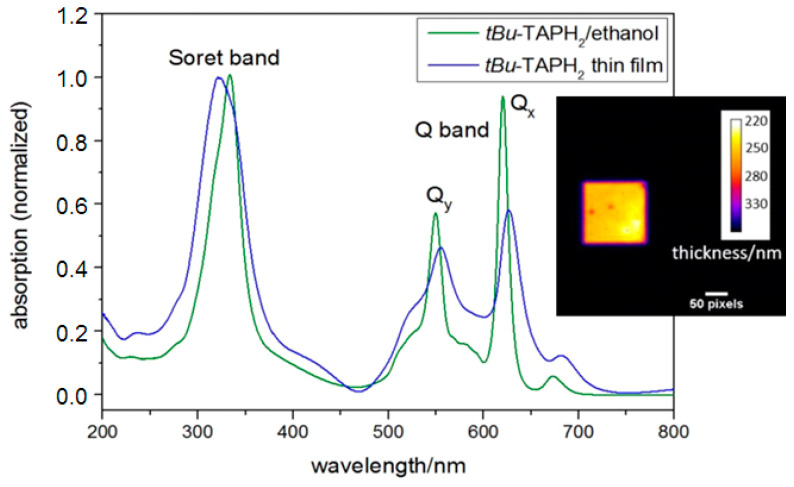
UV/Vis spectrum of TAP in solution and prepared as a thin film (thickness of 120 nm). Inset: EUV transmission image of a thin film TAP sample (thickness about 250 mn).

**Figure 11 ijms-22-13463-f011:**
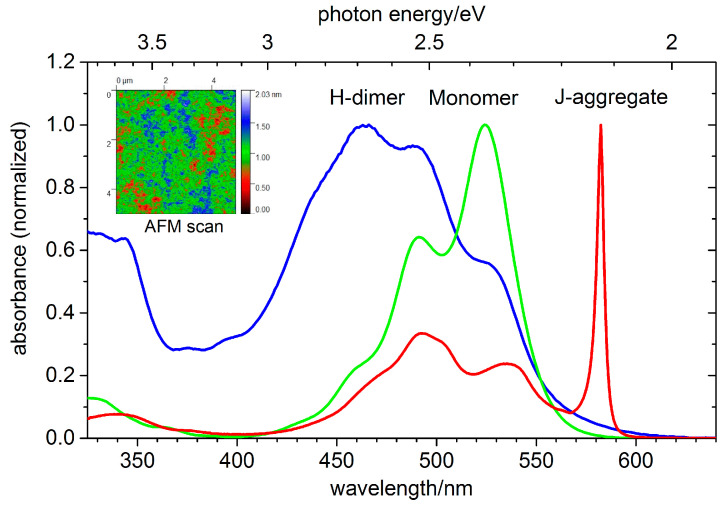
UV/Vis absorption of PIC. The thin film spectrum (blue) shows a blue shift in comparison with the monomer (green curve) spectrum in solution, indicating an H-aggregation of the film. At concentrations > 10^−2^ mol/L, the characteristic J-band appears (red curve) in aqueous solution. inset: AFM analysis of the homogeneity of the prepared thin film (thickness 270 nm).

**Table 1 ijms-22-13463-t001:** Assignment of the measured peak positions for the carbon and nitrogen K-edge NEXAFS spectra of TAP and PIC.

		Carbon K-Edge		Nitrogen K-Edge		
	Feature	Measured (eV)	Assignment	Feature	MEASURED (eV)	Assignment
TAP	A	283.9 ± 0.2	C bound to C (pyrrole)	A′	398.4 ± 0.2	N 1s -> π* LUMO
	B	285.4 ± 0.2	C bound to N (pyrrole)	B′	400.1 ± 0.2	N 1s -> π* LUMO+1
	C	287.4 ± 0.2	C bound to C (pyrrole)	C′	401.5 ± 0.2	N bound to H (pyrrole)
	D	288.2 ± 0.2	C bound to C (butyl)	D′	403.4 ± 0.2	
	E	289.0 ± 0.2	C bound to N (pyrrole)			
PIC	F	285.7 ± 0.2	C bound to C (ring)	F′	398.1 ± 0.2	N 1s -> π* (LUMO)
	G	286.7 ± 0.2	C bound to N	G′	399.8 ± 0.2	N 1s -> LUMO +1
	H	289.3 ± 0.2	C (ethyl)	H′	400.5 ± 0.2	N 1s -> LUMO +3

**Table 2 ijms-22-13463-t002:** Assignment of the measured peak positions for the oxygen K-edge and nickel L-edge NEXAFS spectra of NiO.

Oxygen K-Edge			Nickel L-Edge		
Feature	Measured (eV)	Assignment	Feature	Measured (eV)	Assignment
A	531.6 ± 0.2	Ni 3d—O 2p mixing	A′	852.6 ± 0.2	2p_3/2_—3d
B	537.1 ± 0.2	Ni 4sp	B′	854.3 ± 0.2	2p_3/2_—3d
C	540.0 ± 0.2	Ni 4sp	C′	865.9 ± 0.2	2p–4sp [51]
D	546.0 ± 0.2	Multiple scattering	D′	869.8 ± 0.2 870.6 ± 0.2	2p_1/2_-3d (f_2g_ e_g_) Multiplet splitting
E	555.4 ± 0.2	Multiple scattering	E′	885.4 ± 0.2	Multiple scattering
F	561.2 ± 0.2	Multiple scattering	F′	902.4± 0.2	Multiple scattering

## Data Availability

The authors confirm that the data supporting the findings of this study are available within the article and its Appendix A.

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
