# Peer review of "Towards Understanding Excited-State Properties of Organic Molecules Using Time-Resolved Soft X-ray Absorption Spectroscopy"

_ijms, 2021, doi:10.3390/ijms222413463_

Round 1

Reviewer 1 Report

The manuscript by Stiel et al. reports an experimental approach for elucidating the structure of transient electronic states using ultrafast laboratory based soft X-ray sources. The authors develop two experimental setups for tr-NEXAFS experiments based on a LPP emitting picosecond and sub-nanosecond soft X-ray pulses in photon energy range between 50 and 1500 eV. To determine the practical use of these approaches, the setups are applied for NEXAFS investigations of thin films of two organic molecules (TAP and PIC) at the carbon and nitrogen K-edge and the results are compared with DFT simulations. The authors report enhancement of the spectral range of high resolving power by a factor of more than 30 upon developing the structures on curved substrates facilitating the analysis of multiple edges. In addition, the spectrometer design offers minimal radiation damage with complete background-free system operation. Nevertheless, the optical pumping does lead to heating of the samples, thereby possibly introducing temperature effects which complicate analysis of the transient signals. Towards this, implementation of pre-characterization concerning temperature effects must be carried out separately for each new sample system which the authors have addressed in the conclusion part.

Overall, this is a good and timely research work and I agree that further development of NEXAFS spectroscopy beyond organic films and to other ultrafast lab-based sources should be promising to probe transient structures on an atomic level. However, I do not completely agree with some of the claims made by the authors which need to be further supported by results and explanations. I do not recommend the manuscript in its present form for publication and feel that the authors should address all the comments appropriately before making any further judgement on the manuscript. My comments are listed below.

Major

  1. I feel that the authors should rewrite the abstract highlighting the key results of this work rather than giving descriptions on the scope and advantages of Optical pump soft X-ray probe spectroscopy. The scope, significance and applications of Optical pump soft X-ray probe spectroscopy can be discussed in the introduction section.
  2. Please specify the x- and y-axis as well as scalebar units for Figure 2.
  3. I believe that the correct term should be “vibrational cascade” rather than “vibrational relations” in the Jablonski Diagram Figure 3. Also, the state to which “ISC” process takes place is not visible. What do the grey solid lines in each S1, S2 and SN indicate? Also, the correct representation of fluorescence should be with a colored arrow while that for excitation should be with a solid arrow? Please correct the Jablonski diagram in Figure 3.
  4. Please provide reference for “In porphyrins and its derivatives, the S1 state splits into two states due to molecular symmetry, which are named Qx and Qy. Transitions to the S2 state lead to the formation of the Soret band or B band” in page 7.
  5. In page 7 (under TAP subheading), the authors state the TAP film thickness to be 120 nm which are considered for the UV/Vis measurements. However, in the same section the authors further report the estimated average thickness (inset of Figure 6) to be 250 nm with small inhomogeneities. This is confusing – the offset seems to be huge.
  6. In page 8 (under PIC subheading), the authors state that “In contrast to other works [56] our preparation technique results in the formation of H-aggregates in the film with a hypsochromic shift of the absorption band in comparison with the monomer”. From the UV-Vis spectral wavelengths in Figure 7, I could see a mixture of H-aggregates and monomers existing in the film state. Also, could the authors provide a justification for this contrasting behavior of aggregation where in normal they are prone to form J-aggregates.
  7. How were the samples (thin films) made? Please include a brief section as “methods” addressing the steps adapted for sample preparation.
  8. What is the reason for the enhancement of d2 and d4 peaks? Also, could the authors provide a justification for the greater increase for d4 > d2 > I could not see any concrete evidence addressing that the density of unoccupied states in the d3 region is transferred to d1.

Minor

  1. Abstract – “allows understand the excited state… ” should be “allows understand the excited state..”.
  2. Figure 8 caption – “LUMO extents over” should be “LUMO extends over”.

Reviewer 2 Report

This work described two experimental setups for the transient soft X-ray absorption spectroscopy based on the LPP emitting picosecond and sub-nanosecond soft X-ray pulses in the photon energy range between 50-1500 eV. There exist some issues should be paid the authors’ attention.

  1. The authors selected three samples to study their excited state properties, why did the authors only choose these three samples for testing? Does the author consider adding some more samples for testing in order to obtain more reliable experimental data?
  2. In section 3.5, the author needs to explain whether the ground state optimization is carried out; which the density functional theory (DFT) or time-dependent density functional theory (TD-DFT) does the author use? Necessary references need to be cited.
  3. As we all know, there are four forms of electronic transition. In fig.3, only the electronic transition form of pi-pi* is considered, why not consider the other transition forms, such as σ→σ*, n→σ*, n-pi*?
  4. In fig.4, the molecular structure of NiO is not clear, please revise it. In addition, for NiO, why did the author not simulate its properties in theory.
  5. Please show the prepared 300 nm thick NiO sample in the manuscript.
  6. In fig.8, what did the red and blue regions mean? The present form of display is easily confusing. Please explain it in the manuscript.
  7. In addition to exploring the properties of excited states, can the time-resolved soft X-ray absorption spectroscopy study other properties of compounds? Please indicate in the manuscript.

Round 2

Reviewer 1 Report

The authors have addressed all the comments. No further revision is required. The manuscript can be accepted for publication.

Reviewer 2 Report

The authors have revised this manuscript, which may be considered.